# Randomized controlled evaluation of the psychophysiological effects of social support stress management in healthy women

Nadja Heimgartner*, Sibylle Meier, Stefanie Grolimund, Svetlana Ponti, Silvana Arpagaus, Flurina Kappeler, Jens Gaab

Division of Clinical Psychology and Psychotherapy, Faculty of Psychology, University of Basel, Basel, Switzerland

* nadja.heimgartner@unibas.ch

## Abstract

Considering the high and increasing prevalence of stress, approaches to mitigate stress-related biological processes become a matter of public health. Since supportive social interactions contribute substantially to mental and physical health, we set out to develop a social support stress management intervention and examined its effects on psychophysiological stress responses as well as self-reported stress in healthy women. In a parallel-group randomized controlled trial, registered in the DSRK (DRKS00017427), 53 healthy women were randomly assigned to a social support stress management or a waitlist control condition. All participants underwent a standardized psychosocial stress test where physiological and emotional stress responses were assessed by repeated measurements of cortisol, heart rate, heart rate variability and state anxiety. Also, all participants completed self-report questionnaires of perceived stress and social support at pre-intervention, post-intervention and follow-up four weeks later. Participants in the social support stress management showed a significantly attenuated integrated state anxiety response in comparison to those in the control condition, but conditions did not differ in any of the assessed physiological stress responses. The intervention significantly reduced perceived stress in comparison to the control condition, but perceived stress levels returned to baseline at follow-up. Our results indicated that the intervention had no effect on physiological responses to acute psychosocial stress, even though anxiety responses to stress were attenuated. However, the social support stress management intervention had a significant, albeit transient impact on perceived stress.

## Introduction

While the ability to respond both physiologically and psychologically in the face of adversity is functional, enduring exposure to stress has a negative impact on mental and somatic health, partly mediated through its effects on the hypothalamic pituitary adrenal (HPA) axis as well as on the sympathetic nervous system (SAM) [1]. Accordingly and for example, stress has been associated with the incident of upper respiratory infections [2], exacerbation in autoimmune

**Data Availability Statement:** Due to the Swiss regulation on human research we are not allowed

to publish data from a clinical trial on a public repository. The Department of Clinical Research of the University of Basel set up a data access committee, which publishes data in form of metadata on a public repository and answers to requests from researchers to get access to the original data. The data set consists of the data required to replicate all study findings reported in the article, as well as related metadata and methods, and it will be available to researchers who meet the criteria for access to confidential data upon request to the data access committee of the Department of Clinical Research of the University of Basel. Access requests for the data underlying this study can be sent to Constantin Sluka (constantin.sluka@usb.ch) of the Department of Clinical Research.

**Funding:** The authors received no specific funding for this work.

**Competing interests:** The authors have declared that no competing interests exist.

diseases [3, 4], increased risk for coronary heart disease [5, 6], development of functional abdominal pain disorders [7], slower wound healing [8], and depressive symptoms [9]. Considering the high and increasing prevalence of stress–a survey in the United States in 2012 showed that 22% of respondents stated extreme stress in their daily lives and 44% that their stress had increased in the past 5 years [10]–approaches to mitigate stress-related biological processes become a matter of public health.

Here, a social perspective is warranted and of potential. In their seminal meta-analyis of 208 laboratory studies, Dickerson and Kemeny [11] identified uncontrollable social evaluation as the most potent stressor in terms of magnitude of the cortisol reaction and time to recovery. But while social interactions have the potential to be potent stressors when treathening one's social esteem, acceptance or status [12], social interactions can also contribute substantially to mental and physical health when perceived supportive. For example, Holt-Lundstad et al. [13] reported an 50% increased likelihood of survival for elderly individuals with adequate socials relationships compared to those with poor or insufficient social relationships over an average of 7.5 years. These protective effects of social support can be explained through both modeling or encouraging healthy behavioral and psychological processes [14] as well as buffering behavioral or physiological responses to acute or chronic stressors [e.g. 15–18].

However, interventions intended to manage stress have yet to explicitly encorporate a social perspective as they are predomently based on a intraindividual understanding of stress, i.e. with a focus on individual stress-related cognitions and behaviors [19–23]. To the contrary and to the best of our knowledge, only one study evaluated the effects of an intervention employing the buffering effect of social support, albeit with no effects on an array of psychological parameters and symptoms as well as cardiovascular reactivity to a psychosocial stressor [24].

Thus, considering its protective and buffering effects, it seems promising to make use of social support in stress management interventions. Therefore, we set out to conceptualize, implement and evaluate the effects of an intervention intended to employ and improve social support skills and to reinforce the stress buffering effect of social support on psychoneuroendocrine stress responses.

## Materials and methods

A parallel-group randomized controlled trial with an intervention and a waitlist control condition was conducted between February 2015 (beginning of the recruitment of the first trial) and December 2015 (follow-up measurements of the second trial) at the University of Basel, Switzerland. The study was carried out in accordance with the Declaration of Helsinki principles and the local ethic committee (Ethikkommission Nordwest- und Zentralschweiz) approved the study protocol and informed consent (reference number EKNZ 2015–005) on 12. January 2015. It is registered in the DSRK (Deutsches Register Klinischer Studien) and listed in the International Clinical Trials Registry Platform of the WHO (ID DRKS00017427). Registration of the study was delayed (14.6.2019) and performed after the termination of the study as the registration of trials on healthy subjects was not standard procedure in 2015, i.e. when the study was run. The authors confirm that all ongoing and related trials for this intervention are registered.

### Participants

Eligible participants were German-speaking healthy women aged between 18 and 60. We included only women based on the assumption of gender differences in stress responses [25], i.e. that women predominantly engage in a tend and befriend response to stress. Thus, we

expected women to readily accept the rational of our intervention, which would facilitate the implementation of our social support stress management. To minimize the interference of medical conditions and behaviors on stress reactivity and to ensure the safety of participants, women were excluded when they reported current or chronic mental or physical disease assessed on self-report, currently received any medication, were in psychological or psychiatric treatment, and when smoking more than five cigarettes a day. Also and to prevent habituation effects to the social stress protocol, women with previous experiences with the Trier Social Stress Test [26] were excluded. Written informed consent was obtained from all participants.

We expected a medium effect size as previous randomized controlled studies on stress management interventions have demonstrated medium to large effect sizes for salivary cortisol stress reactivity [19–21]. Thus, estimating an intermediate to large effect size of $f$ = 0.35, G*Power 3.1 calculated a total of N = 48 participants needed to detect interaction effects between two conditions and three measurements at 85% power using the originally planned MANCOVA for repeated measures [27].

## Procedure

In February/March 2015 (trial 1) and September/October 2015 (trial 2) participants were recruited in lectures and via online advertisement at the University of Basel and the University of Applied Sciences and Arts of Northwestern Switzerland. Interested women completed an online screening questionnaire. A total of 53 women fulfilled inclusion criteria and were randomly allocated by two graduate students (SG and FK) in a 1:1 ratio to the intervention or the waitlist control condition, with allocation concealment by opaque sequentially numbered sealed envelopes. As we planned to include more subjects in trial 1 than we could finally engage and opaque envelopes with allocation to condition were prepared in advance, not all prepared envelopes could be used, resulting in a uneven distribution of participants to the two conditions. Participants in the intervention condition received a two-week social support stress management (SSSM) in groups of 5–8 participants (week 2 and 3). In the waitlist control condition, participants received the intervention after study completion (after week 8). Self-report questionnaires were completed online at baseline (week 1), post-intervention (week 4) and follow-up (week 8). Between the post and the follow-up assessment (week 5 to 7) all participants underwent a psychosocial stress test with assessement of neuroendocrine, cardiovascular and psychological stress responses. The two confederates leading the psychosocial stress test and data analysts were blinded to condition assignment. Data collection was to 62% completed by study conductors blinded to condition assignment. Blinding of participants and intervention providers was not possible due to the study design.

## Intervention

The SSSM was conceptualized to address the importance of social support for health and well-being and to improve interpersonal skills to give and receive social support. Besides providing information on social support effects and its possible mechanisms, the SSSM focused on modeling supportive, non-judgemental communication between therapists and participants and on carrying out exercises in small groups to improve interpersonal communication skills. With this, the aim was to enhance the quality of interpersonal interactions and to allow personal and interpersonal exploration. Participants were encouraged to practise these skills in real life, i.e. between sessions, and to actively provide as well as ask for social support, since bi-directional support seems to be more effective in terms of well-being than just receiving support [28]. The intervention was based on a manual (for details, see below) and adherence to the manual was supervised by a certified psychotherapist (JG).

The intervention consisted of 16 training hours divided into 6 sessions with the first and the final session lasting 4 hours and the other four sessions 2 hours each. Each session focused on a different aspect of social support (see S1 Appendix). All sessions took place within a two-week period. Two graduate students in clinical psychology (SG, SP) and a PhD student (NH) with substantial training and experience in psychotherapy conducted the SSSM in the intervention condition. In the control condition, the intervention was conducted by four graduate students (SG, SP, SA, FK) and a PhD student (NH). Graduate students were constantly supervised by the PhD student and all therapists were supervised weekly by a certified psychotherapist (JG).

## Manual of the social support stress management

The goal of the intervention is to impart interpersonal skills to deal with stress and give participants opportunities to train these skills. Most likely these skills are not completely new and at least partially already available. It is important to stress their importance, to implement and practise them out in a safe environment under professional supervision. Therefore, therapists should model social support by communicating openly and empathically and to validate each other and participants. Problems should be approached in a solution-focused way with emphasis on what was, is or could be helpful. Topics should be discussed first on the basis of personal experience and then in the context of scientific studies and models. At the end of each session, participants should be encouraged to share their experiences of the session, verbalize what they take home from the session and what they could try out in the next days.

The goal of the first session (4 hours) is to introduce and discuss the aim of the intervention and to train basic communication skills. In the first module (60 minutes), participants discuss what puts them under stress and what helps them to cope with stress. Therapists seek to put the results of the discussion in context of findings on social evaluative threat and the social dimension of stress [11], on social support [13] and the assumption of gender-specific coping [25]. In the second module (90 minutes), participants and therapists play a board game involving personal questions. The goal of the game is to create a relaxed atmosphere where participants and therapists can get to know each other. In the third module (90 minutes), participants are invited to discuss and practice determinants of good communication, such as empathy, authenticity and unconditional appreciation.

The goal of the second session (2 hours) is to improve the perception and communication of one's own feelings [29] as a basis for supportive communication. For this purpose, participants are introduced to simple exercises to monitor somatic representations of emotional experiences, which are first demonstrated to the whole group under the guidance of a therapist and then exercised in small groups.

The goal of the third session (2 hours) is to focus on one's own social network and the different forms of social support that are enacted in this network, especially when under stress [30]. Participants discuss different forms of support and draw an illustration of their social network. They discuss how their social ties are affected by stress and how and with whom they can increase beneficial social interactions when under stress.

The goal of the fourth session (2 hours) is to address self-disclosure and being vulnerable in social interactions in the delicate balance between feeling threatened by anticipated and real social rejection and the relieving experience to share problematic experience with others. These topics are discussed on the basis of personal experiences and with references to Leary's sociometer theory [31].

The goal of the fifth session (2 hours) is to differentiate between different types and consequences of stress and how to utilize social support when under stress. In the first module (60

minutes), participants and therapists discuss differences between acute and chronic as well as functional and health-impacting stress. In the second module (60 minutes), the use of imagined persons as social support in times when no-one is directly available is discussed and experiences with this are shared with the group.

The goal of the sixth session (4 hours) is, beside the presentation of one new aspect of social support, the review and repetition of contents of the intervention. In the first module (60 minutes), the relevance and importance of physical contact as a form of social support is discussed on the basis of personal experience and scientific research [15]. In the second module (30 minutes), participants discuss what was important for them and possible changes they made during the intervention in small groups. In the closing module (90 minutes), the game from the first session is played again, this time with questions focused on experiences made during the intervention. A feedback round followed by a small reception with snacks and drinks concludes the intervention (60 minutes).

## Stress test

The Trier Social Stress Test (TSST, [26]) was used to induce psychosocial stress. The TSST consists of a simulated job interview followed by a mental arithmetic task (5 minutes each) in front of an audience of two confederates. The TSST has repeatedly been found to induce profound endocrine and cardiovascular responses in 70–80% of the subjects tested [26]. Neuroendocrine, physiological and psychological data were collected before, during and after the test (for details see below). To account for circadian rhythm in cortisol secretion [32], all TSST took place between 1:55pm and 6:00pm (mean = 3:58pm; no differences between conditions: $F(1,35) = 0.02$, $p = 0.91$). The appointment was on average 23.55 days (SD = 6.99) after the last day of the intervention (range: 16–37 days, with 75% of appointments occurring within 26 days) in the intervention condition. After completion of data collection, participants were fully debriefed on the nature of the TSST and compensated with 50 CHF or study credits.

## Measures

Demographic data including age and body mass index (BMI) were assessed online together with the inclusion criteria before condition randomization took place. At baseline (week 1), post-intervention (week 4) and at follow-up (week 8) the following questionnaires were administered online:

The German version of the Perceived Stress Scale [PSS; 33], consisting of 14 items rated on a 5-point scale, was used to assess the degree to which situations in the last few days have been appraised as stressful. PSS scores can be in the range between 0 and 56 with higher values indicating more perceived stress. In a probability sample of the United States, Cohen [34] found mean scores for women of 20.2 (SD 7.8).

The German version of the State Trait Anxiety Inventory [STAI; 35] was used to assess trait anxiety. The STAI trait form consists of 20 statements each to be rated on a 4-point scale with a total score of 20–37 interpreted as little or no anxiety, 38–44 as moderate anxiety and a score of 45–80 as extreme anxiety.

A German change-sensitive symptom list [ASS-SYM; 36] was employed to assess experience of relaxation, well-being, discomfort and preoccupation. The ASS-SYM has 6 subscales (physical and psychological exhaustion, nervousness and inner tension, psychophysiological deregulation, performance and behavioral problems, burden of pain, problems with self-determination and -control) with 8 items each, rated on a 4-point scale. Additionally a sum scale, which can have scores between 0 and 192, is calculated to reflect the general level of symptoms and problems. Norms for a heterogenous German sample showed values between 26 and 55 for the 34th to 66th percentile.

The availability of social resources was assessed with the short version of the German Social Support Questionnaire [FSozU K-22; 37]. The FSozU K-22 includes 22 statements about the perceived and anticipated availability of social resources to be rated on a 5-point likert scale. It covers the dimensions emotional and practical support and social integration. A score is calculated as mean of the sum of all items and can have values between 1 and 5. Norms for a representative German sample showed a mean of 4 (SD 0.66) and values from 3.69 to 4.41 for the 30th to 68th percentile.

At post-intervention assessment, two questionnaires were administered to the intervention condition to evaluate perceived group climate and therapeutic alliance:

The German version of the Group Climate Questionnaire—Short Form [GCQ-S; 38] was employed to assess perceived engagement (i.e. group cohesion, cognitive understanding, self-disclosure and empathy) and conflict (i.e. anger, detachment, confrontation and mistrust) with 4 items each. Both scales are calculated as mean values of the four corresponding items and can have values between 0 and 6.

The German version of the Working Alliance Inventory—short revised [WAI-SR; 39] was employed to asses the therapeutic alliance, i.e. agreement on tasks, agreement on goals and development of an affective bond. The 12 items of the WAI-SR were rated on a 5-point scale from 1 to 5. The score represents the mean of the 12 items and can have a value between 1 and 5. The wording of the questionnaire was adapted to fit our intervention [e.g. 40].

Furthermore, to assess physiological and emotional responses during and to the TSST, the following parameters were assessed:

To assess state anxiety before and after the TSST, the German version of the State Trait anxiety inventory [STAI; 35] was administered before the TSST (-45 minutes), after the introduction to the TSST (-20 minutes), immediately after the TSST (0 minutes) and in the recovery phase (50 minutes).

To assess salivary free cortisol levels, nine saliva samples were collected using Salivette collection devices (Sarstedt, Sevelen, Switzerland) at -45, -35, -25, -10, 0, 10, 20, 35 and 50 minutes. Sampling time lasted approximately 1 minute during which subjects chewed on the cotton swabs as regularly as possible. Salivettes were stored at −20˚C until biochemical analysis took place. After thawing, biochemical analyses were conducted in the bio-chemical laboratory of the Clinical Psychology and Psychotherapy department at the University of Zurich, Switzerland, by means of a highly sensitive liquid chromatography–tandem mass spectrometry (LC–MS/MS) method [41]. Since the use of oral contraceptives and the menstrual cycle phase has been shown to influence the activity of the HPA-axis and therefore the endocrine response to stress [42], participants reported the use of oral contraceptives, menstrual cycle length and the first day of their last menses. Based on these informations, menstrual cycle phase, i.e. follicular or luteal phase, on the day of the TSST was estimated. The follicular phase was defined as the period between the first day and 14 days before the end of the menstrual cycle, while the luteal phase was defined as the last 14 days of the cycle [43].

Electrocardiography (ECG) was recorded continuously at 1000 Hz during 1 hour and 40 minutes using the wireless physiological recording system BioNomadix® (Biopac Systems, Inc., Goleta, CA). Recorded ECG data were filtered using the software AcqKnowledge (Biopac Systems, Inc., Goleta, CA) with a FIR bandpass filter from 0.5 Hz to 35 Hz with 8000 coefficients. The resulting heart period series were visually examined for artifacts and corrected when necessary using the VivoSense® software (Vivonoetics, Inc., San Diego, CA). To assess changes in heart rate over the course of the experiment 7 mean heart rate slots of 5 minutes each were calculated. Slots represented the mean heart rate at baseline (-30 to -25 minutes), in the preparation phase of the TSST, during the free speech in the TSST, during the mental arithmetic task in the TSST and in the recovery phase with three slots starting 5, 15 and 25 minutes

after the TSST ended. Furthermore, heart rate variability was calculated as RMSSD, the square root of the mean of the sum of the squares of differences between adjacent normal-to-normal intervals. This parameter is one of the most commonly used measure derived from interval differences [44]. Shaffer and Ginsberg [45] presented norms for short-term measurements (around 5 minutes) of RMSSD with a range of 19 to 75 and a mean of 42 milliseconds (SD 15).

Deviating from the study protocol we did not perform assessments of alpha-amylase and electromyography. Data of electrodermal activity, skin temperature, and respiratory activity were not analyzed. Analysis of two questionnaires (PASA and BFW/E) were not reported.

## Statistical analysis

All calculations were carried out using the statistic software IBM® SPSS® Statistics Version 27. Two sample t-tests were calculated to compare demographic characteristics and baseline values. Associations of age with psychometric or physiological measures were calculated as Pearson correlation coefficients.

For the comparision of psychological and physiological variables between the two conditions over the course of the study and the TSST (condition by time interaction), separate covariance pattern models were employed using the mixed procedure of SPSS. Covariance pattern models allow to deal with missing data and the correlation between repeated measures and were therefore used instead of the previous planned MANCOVA. Normal distribution and homoskedasticity of the residuals were visually inspected and tested using the Kolmogorov-Smirnov and the Levene's test. When normality and homoskedasticity were not met, calculations were repeated with Box-cox transformed data. For better readability, the original values were used for tables and figures.

Covariance pattern models for each outcome were built in the following order: 1) A basic model (model 1) was built including time, condition (i.e. intervention and control), and the interaction of condition by time as fixed effects and time as a repeated effect on the level of the participant. 2) As participants were randomly assigned to one of the two conditions, identical baseline values were assumed for the psychological variables at pre-intervention (t0). Therefore, a model 2 was built for psychological variables including post(t1), follow-up time(t2), the interaction of condition and t1, and the interaction of condition and t2 as fixed effects (based on [46], p. 128 ff). As in model 1, time was defined as repeated effect. 3) Effects of the possible covariates age or BMI were tested by adding them to the model 1 or 2 as fixed effects and comparing model fit of the models including age or BMI to model fit without them. 4) To test the primary interest, i.e. differences of the two conditions over the course of the measurements, a simplified model including the same predictors as model 1 or 2 except the interaction of condition by time was built and model fit was compared to model 1 or 2. When there was a significant model fit improvement in the model with the interaction effect, estimations of unstandardized regression coefficients were reported. The effect size Cohen's d was calculated by dividing the unstandardized regression coefficient with pooled within-group standard deviation of the outcome [47].

All models employed maximum likelihood estimation. Model fit differences were tested with the likelihood ratio test. To achieve optimal fit whilst reducing the parameters, the covariance structure of the repeated effect was set to compound symmetry in models for psychological variables measured pre-, post and follow-up and to first-order ante dependence for psychophysiological variables measured before, during and after the TSST. Due to bad ECG data quality ten subjects had artifact corrections between 3 and 10%. Heart rate and RMSSD analyses were repeated with these 10 subjects excluded, but results did not differ (data are not shown).

For cortisol and STAI state responses areas under the response curve were calculated with respect to increase (AUCi) using the trapezoidal method as an indicator for the integrated response in the TSST [48]. As we were interested in the reaction of cortisol to the TSST, the first three cortisol measurements which took place before the introduction of the TSST were simplified to one baseline measure by calculating their mean. There were two values missing in assessments of the STAI state due to data transfer interruptions which were replaced by multiple imputations (20 iterations). Pooled parameters for the testing of condition differences in AUCi STAI state were reported.

Distribution between conditions in use of hormonal contraceptives, menstrual cycle phase and frequency of non-responders were tested with Chi-square tests of independence. Descriptive statistics are presented for the GCQ and WAI measures.

For all analyses, significance level was set at α = 5%.

## Results

53 women were randomly assigned to the intervention ($N$ = 28) and control condition ($N$ = 25). Two women in the intervention and four women in the control condition withdraw their participation before the first assessment started, so they were not included in the analysis. Five women in the intervention condition did attend the intervention only six hours or less. To be able to test for effects of the intervention, these subjects were excluded from analysis. The mean intervention attendance of the 21 subjects with sufficient intervention attendance was 12.38 hours ($SD$ = 2.58). The sample to analyze the pre/post and follow-up-questionnaires consisted of $N$ = 21 subjects in the intervention condition and $N$ = 21 subjects in the control condition (for details see Fig 1). $N$ = 18 participants in the intervention and $N$ = 16 participants in the control condition completed all assessments including the follow-up assessment. With four participants it was not possible to schedule a date for the TSST within the designated time (week 5 to 7). One participant did not continue assessments and did not respond to mails after the pre-intervention assessment. This resulted in $N$ = 37 participants completing the TSST and $N$ = 37 in the analyis of STAI state and cortisol. Two values in the assessment of STAI state were missing due to interruptions in data transfer. For two participants the recording of the ECG did not work, resulting in $N$ = 35 in the analysis of heart rate parameters. Because of bad recording quality seven values were missing in heart rate and RMMSD data.

The two conditions differed significantly in mean age with participants in the intervention condition significantly younger than in the control condition (intervention condition $M$ = 24.57, $SD$ = 7.44; control condition $M$ = 34.95, $SD$ = 13.87, $t(30.64)$ = -3.02, $p$ = .005, $d$ = .93, *95% CI* [0.29; 1.57]). This difference was caused by an accidental uneven distribution of older participants to the control condition: Nine participants of the whole sample were older than 40 and seven of them were allocated to the control condition. There were no differences between conditions in BMI (intervention condition $M$ = 21.25, $SD$ = 1.86; control condition $M$ = 21.22, $SD$ = 3.12, $t(40)$ = .03, $p$ = .98) and questionnaire scores at baseline (see Table 1). In the TSST, conditions did not differ in baseline cortisol (intervention condition $M$ = 4.68, $SD$ = 3.43; control condition $M$ = 4.74, $SD$ = 2.98, $t(35)$ = -.06, $p$ = .95), heart rate (intervention condition $M$ = 74.92, $SD$ = 10.13; control condition $M$ = 72.93, $SD$ = 12.71, $t(32)$ = .51, $p$ = .62), RMSSD (intervention condition $M$ = 50.94, $SD$ = 24.92; control condition $M$ = 40.83, $SD$ = 22.66, $t(32)$ = 1.23, $p$ = .23) and state anxiety level (STAI state; intervention condition $M$ = 38.15, $SD$ = 8.50; control condition $M$ = 35.53, $SD$ = 5.16, $t(35)$ = 1.11, $p$ = .28). Use of oral contraceptives (intervention condition N = 10/50%; control condition N = 4/24%, $\chi2 (1)$ = 2.74, $p$ = .10) as well as number of participants in the follicular phase (intervention condition 8/40%; control condition N = 6/35%, $\chi2 (1)$ = .09, $p$ = .77) did not differ significantly between conditions.

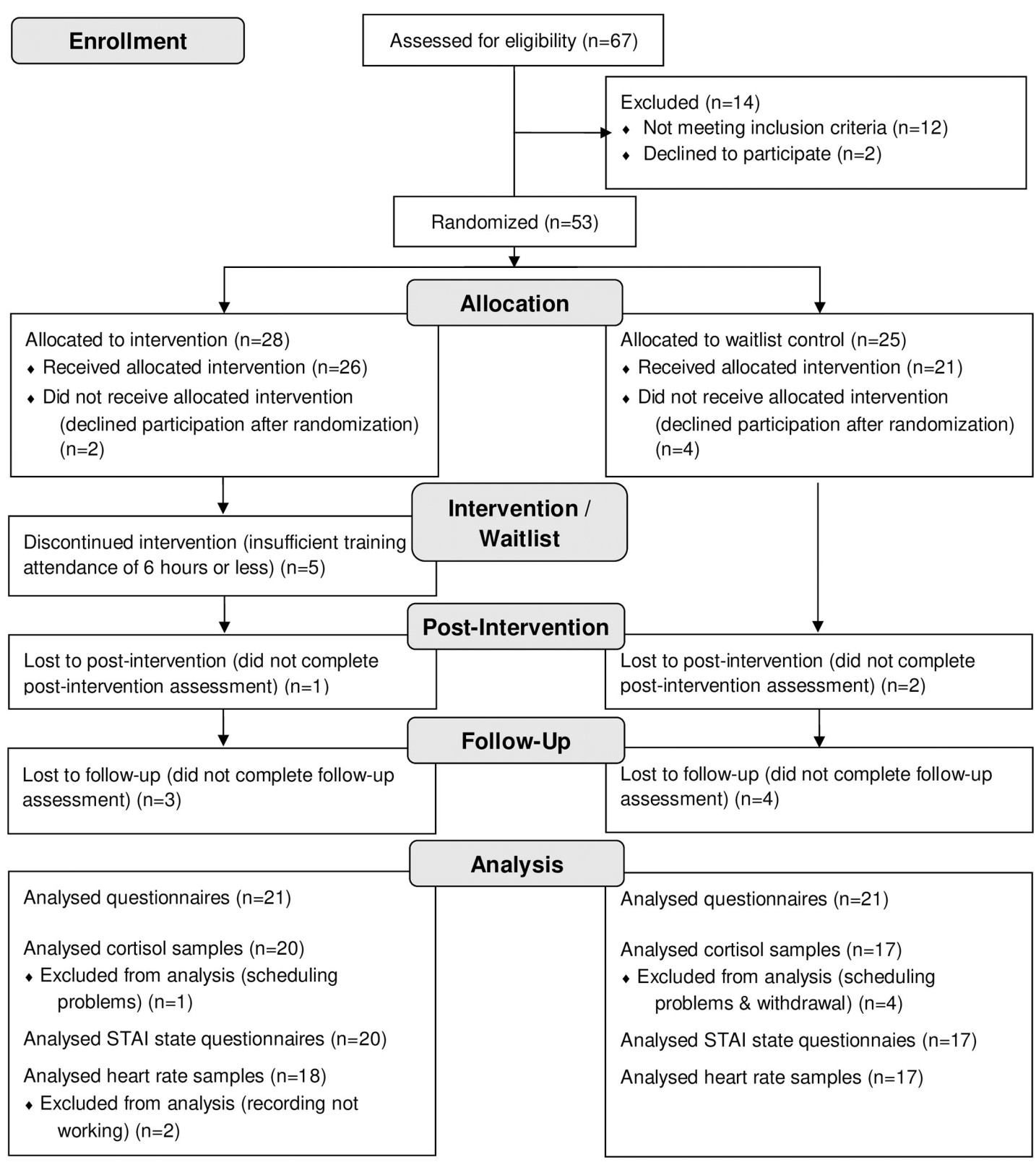

**Fig 1. Recruitment and participants flow through the study.**

**Table 1. Condition differences in psychological variables over the course of the study.**

| | Intervention condition | | | Control condition | | | Intervention vs. Control at Baseline | | |
|---|---|---|---|---|---|---|---|---|---|
| | Baseline N = 21 | Post N = 20 | Follow-up N = 18 | Baseline N = 21 | Post N = 19 | Follow-up N = 17 | Difference of means | 95% CI | T-test |
| PSS | 24.05 (6.35) | 19.35 (5.51) | 23.00 (7.93) | 20.57 (8.94) | 20.79 (9.53) | 21.06 (8.65) N = 18 | 3.48 | [-1.36; 8.31] | $t(40) = 1.54, p = .15$ |
| STAI trait | 37.67 (7.46) | 36.95 (8.07) | 37.67 (9.62) | 35.76 (9.10) | 36.37 (10.47) | 37.12 (9.16) | 1.90 | [-3.28; 7.09] | $t(40) = .74, p = .46$ |
| ASS-SYM | 36.10 (17.56) | 32.75 (16.40) | 37.44 (21.76) | 33.00 (20.23) | 30.05 (21.21) | 34.76 (20.35) | 3.10 | [-8.72; 14.91] | $t(40) = .53, p = .60$ |
| FSozU K-22 | 4.56 (0.32) | 4.57 (0.33) | 4.60 (0.23) | 4.39 (0.49) | 4.32 (0.55) | 4.25 (0.58) | .17 | [-.08; .43] | $t(40) = 1.36, p = .18$ |

Values are given as means ± SD. PSS, Perceived Stress Scale; STAI trait, State Trait Anxiety Inventory Trait Scale; ASS-SYM, Change-sensitive Symptom List; FSozU K-22, Social Support Questionnaire Short Version.

### Psychophysiological stress responses

Cortisol responses over time did not differ significantly between conditions, shown by no significant improvement of model fit when the interaction condition by time was added to the model (model included BMI as covariate, $\chi 2 (6) = 9.80, p = .13$, Box-Cox transformed values $\chi 2 (6) = 8.25, p = .22$; see Fig 2) and no differences between conditions in the integrated cortisol response AuCi: intervention condition $M = 35.50, SD = 199.81$; control condition $M = 40.39, SD = 169.81, F(1, 34) = .004, p = .95$). Conditions did not differ in the prevalence of cortisol non-responders (cortisol increase to baseline < 1 nmol/l: control condition N = 7/41%; intervention condition N = 9/45%, $\chi 2 (1) = .06, p = .82$). Also, there was no significant condition by time interaction effect in the model for heart rate (model included BMI, $\chi 2 (6) = 3.98, p = .68$, Box-Cox transformed values $\chi 2 (6) = 2.30, p = .89$) or RMSSD responses to the TSST (model included age, $\chi 2 (6) = 8.22, p = .22$, Box-Cox transformed values $\chi 2 (6) = 7.28, p = .30$; see Fig 3). Regarding STAI state anxiety responses to the TSST, there was no significant improvement in model fit when the interaction condition by time was added (model included age, $\chi 2 (3) = 4.03, p = .26$; Box-Cox transformed values $\chi 2 (3) = 3.89, p = .27$;); see Fig 2). However, participants in the intervention condition showed a significant lower integrated response

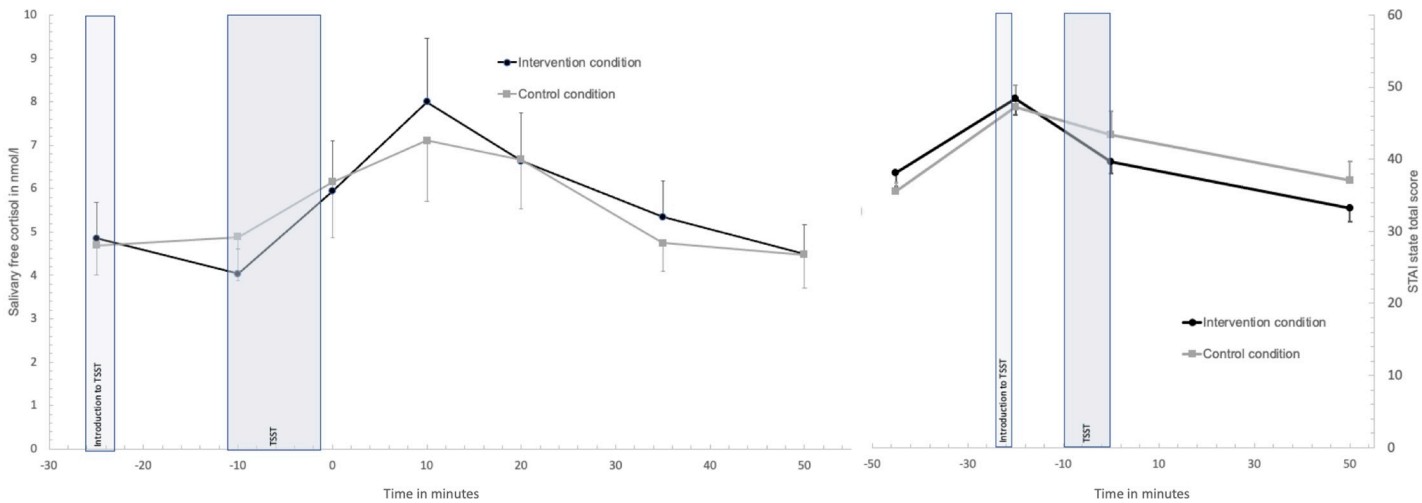

**Fig 2. Salivary free cortisol and state anxiety responses in the Trier Social Stress Test between conditions.** Values are means and standard error means.

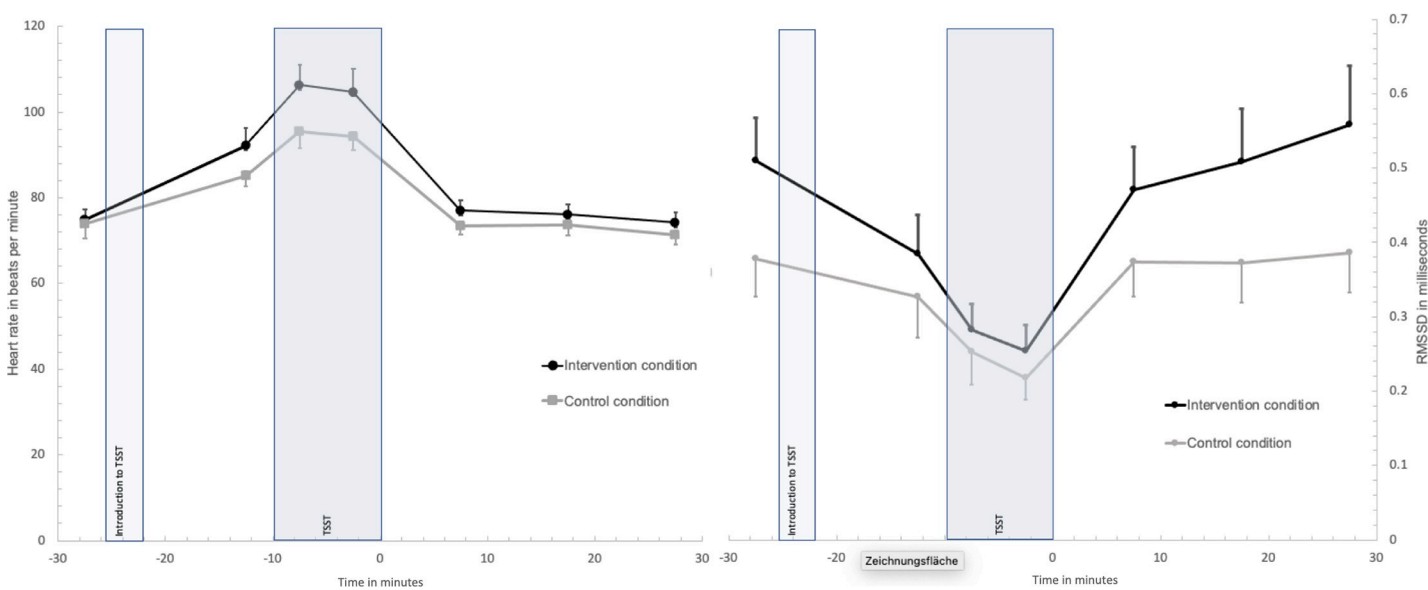

**Fig 3. Heart rate and RMSSD responses in the Trier Social Stress Test between conditions.** Values are means and standard error means.

in state anxiety in comparison to controls (pooled results from multiple imputations for two missing values: AuCi intervention condition $M = 191.71$, $SEM = 120.13$; control condition $M = 619.91$, $SEM = 217.81$, $F(1, 34) = 4.38$, $p = .045$, partial $\eta^2 = .11$; original data: AuCi intervention condition $M = 153.21$, $SEM = 118.21$; control condition $M = 592.63$, $SEM = 227.55$, $F(1, 32) = 5.51$, $p = .025$, partial $\eta^2 = .15$). All variables measured over the course of the TSST showed significant changes over time (fixed effect of time: cortisol $F(6, 78.56) = 4.41$, $p = .001$; heart rate $F(6, 62.11) = 24.32$, $p < .001$; RMSSD $F(6, 60.89) = 19.98$, $p < .001$; STAI state $F(3, 51.41) = 24.92$, $p < .001$).

## Effects on perceived stress

Comparing model 2 with and without interaction of condition by time showed a significant improvement of model fit for PSS when the interaction was included (model included age, $\chi2 (1) = 7.91$, $p = .005$). The effect of condition by time was significant at post-intervention with the intervention condition showing a decline in PSS values ($b = -4.70$, 95%CI [-8.43; -.97], $F(1, 95.97) = 6.26$, $p = .014$, $d = -.60$) but not at follow-up ($b = .90$, 95%CI [-2.96; 4.76], $F(1, 95.34) = .22$, $p = .64$). The analysis was repeated with no restrictions regarding minimal attendance of the intervention and did show a less pronounced but still significant effect for model fit improvement when the interaction effect was added ($N = 47$, $\chi2 (1) = 4.66$, $p = .031$, condition by time at post-intervention: $b = -4.08$, 95%CI [-7.94; .21], $F(1, 100.03) = 4.38$, $p = .039$; condition by time at follow-up: $b = .46$, 95%CI [-3.52; 4.45], $F(1, 99.33) = .05$, $p = .82$).

There was no indication for differences in conditions over the course of the study in any other psychological variables, as the inclusion of the condition by time interaction did not lead to a significant improvement in model fit (STAI trait: $\chi2 (1) = .66$, $p = .42$; ASS-SYM: $\chi2 (1) = 1.32$, $p = .25$; FSozU K-22, model with age: $\chi2 (1) = 2.94$, $p = .09$; see Table 1).

## Evaluation of the SSSM

Results of the GCQ-S indiated high levels of engagement ($M = 4.93$, $SD = 0.62$) and very little interpersonal conflict ($M = 0.49$, $SD = 0.63$). Furthermore, the overall alliance (WAI-SR) with

therapists ($M = 3.40$, $SD = 0.58$) and participants in the SSSM condition ($M = 3.41$, $SD = 0.54$) was good and comparable. On the three subscales, development of an affective bond was rated high ($M = 4.05$, $SD = 0.58$ for therapists, $M = 3.98$, $SD = 0.34$ for participants), ratings for agreement on tasks and agreement on goals were one point lower on the likert scale (task $M = 3.08$, $SD = 0.87$ for therapists, $M = 3.18$, $SD = 0.84$ for participants, goal $M = 3.09$, $SD = 0.70$ for therapists, $M = 3.08$, $SD = 0.81$ for participants).

## Discussion

The goal of the study was to implement and evaluate a social support stress management intended to employ and improve social support skills and to reinforce the stress buffering effect of social support on psychological parameters and psychophysiological stress responses in healthy women. The intervention was favourably rated by participants and led to a significat–albeit transient–reduction in perceived stress. Although some improvements in the availability of social resources, levels of trait anxiety, relaxation and preoccupation were reported after the intervention, none of them were significant when compared to the control condition. Importantly, the intervention had no effects on cortisol, heart rate and heart rate variability responses to the standardized psychosocial stress test, although participants in the intervention condition showed a significantly attenuated state anxiety response in comparison to those in the control condition, but only when the integrated state anxiety response was compared.

The newly conceptualized social support stress management intervention was perceived as practicable in terms of length and effort as well as generally rated very positive, with high engagement and low conflict ratings, at least when compared with group psychotherapy [49, 50]. Also, ratings of bond with psychologists and participants in the intervention condition were high and comparable to those found in other studies [51]. However, ratings of agreement on tasks and goals are rather low, possibly due to the focus of the intervention laying on supportive interaction and not on defining individual tasks and goals.

The intervention led to increased levels of perceived social support whereas participants of the control condition showed a decrease in this measure, but this observation was rather moderate and not significant. This differs from trials testing other interventions to strengthen social support, which found trends [24] or significant improvements in social support measures [52, 53]. However, although these studies shared some similarities with our study, such as inclusion of healthy participants and the aim to improve and practise social skills, these studies only included individuals scoring low on social support measures and also included male participants. In contrast, our sample had already high baseline ratings in perceived social support, with scores 12% above those reported in a normative sample of healthy subjects [37]. Thus, the lack of any effect on social support measures in our participants could be seen as a consequence of already high social support levels at baseline. However, it is also possible to question whether social support as such is amenable to training. This is further substantiated by the transient nature of the observed effects on perceived stress. Thus it is possible that the effects were caused by the perceived social support during the intervention, but not through increased abilities to obtain social support. Further studies are needed to examine possible and sustainable training effects of similar trainings on social support skills.

With regard to perceived stress, the social support stress management led to a significant reduction of perceived stress of medium effect size, but which also returned to baseline levels at follow-up. These findings partly correspond to previous evaluations of stress management trainings, which similarly reported reduced levels of perceived stress after the training, but were based on different theoretical premises [19, 21]. However, not all interventions based on intraindividual stress management could find an effect of the intervention on perceived stress

[23] and also trainings intended to improve social support so far failed to show an effect on daily stress or perceived work stress [24, 54]. Assumably, our intervention helped participants to reduce stress by giving the opportunity to interact with each other in a supportive way during the intervention, but this did not lead to lasting changes in behavior and transfer into daily life. A comparision with other interventions improving social support skills is difficult, as most testing did not inlcude a follow-up. Of the 13 studies reviewed [55] on social support skills group trainings, only 4 of 13 studies inlcuded a follow-up, of which three reported effects to be maintained at follow-up three or six months later [56–58]. It has to be taken into account that the studies showing long-term effects were conducted on a population with psychiatric disorders and having considerable deficits in social skills. To the best of our knowledge, only one study found positive effects of a social support training maintained at the follow-up 10 weeks later in non-clinical subjects, which, however, scored low on social support measures [53]. It is possible that individuals with considerable deficits in social skills and social support would profit from similar trainings and be able to transfer trained skills into their everyday life.

The TSST elicited an increase in cortisol, heart rate and state anxiety and a decrease in RMSSD. But while heart rate responses to the TSST were comparable to those found in other studies [15, 59], cortisol responses have to be considered as lower than norm [i.e. with increases of 50% to 150%; 60] as 16 subjects of our sample were cortisol non-responders. This is possibly due to the inclusion of women on oral contraceptives and in the follicular phase of the menstrual cycle [42] and that our participants, of which more than half were students, were used to hold presentations or speak in public [61].

When comparing the intervention to the control condition, there were no differences in their physiological reaction to the TSST, indicating no effect of the intervention on acute stress response. While some of previous stress management trainings with a clear intra-individual focus found effects on the cortisol response to acute stress [19–21], others did not [23] as well as trainings based on social support failed to find effects on the cardiovascular stress response [24, 54]. With regard to our intervention, this finding is possibly due to the protocol of the TSST, which clearly prevents any possibility to use or obtain social support from present persons, as the audience of confederates is instructed to interact in a neutral way with the participants and not to give any positive verbal or non-verbal feedback or signs of communication other than that specified in the manual [26]. All efforts of participants to interact with members of the audience in possible supportive ways were therefore unsucessful and otherwise stress-reducing social skills could not be applied. To test the potential of similar social support stress managements to reduce psychophysiological responses in stressful situations, the latter should strive for ecological validity and thus offer opportunities for positive interactions, as many studies indicate that positive social interactions before or during a psychosocial stress test can reduce cardiovascular and endocrine reactivity to the stressor [15, 62–66]. Besides receiving social support from present persons, the possiblity to access internalized social support should be addressed. In session five of the intervention, participants were instructed to think of a imagined person to receive social support from in times when no one is directly available. Jakubiak and Feeney [67] found lower stress ratings in the TSST for participants who imagined touch support from a romantic partner than participants imagining verbal support or a control imagination task. In our study we did not assess if participants used the support of a imagined person during the stress task. Therefore, no conclusions can be drawn if the application of imagined support was not helpful enough to buffer the physiological stress response or if participants simply did not think of the possibility to imagine a supportive person.

Also, as participants were not blind with regard to their allocation, it is possible that the observed intervention effects on perceived stress are caused or influenced by the knowledge

that there are receiving an intervention and that this intervention is thought to be effective. Response expectancy is a known mediator of treatment responses in both pharmacologial as well as psychological interventions [68, 69] and thus, the knowledge to be allocated to an intervention might have triggered the observes effects on perceived stress. However, as expectancy is part of psychological interventions per se and blinding is methodologically near impossible [70], the same could be said about any psychological intervention.

The following limitations have to be considered when interpreting the results of our study. Unintendedly, there was a significant difference in age between the two conditions, which was accounted for by controlling for effects of age in all calculations. The sample size was rather small, limiting the statistical power to detect small condition differences. Because of drop out after randomization we did not reach our intended N of 48 participants and therefore can't exclude that the study was underpowered to detect a significant effect of the intervention on the physiological stress reaction. However, our data do not indicate an attenuated stress reaction after the intervention in any of the physiological variables and we do not expect that there would have been completely different results with a bigger sample size. To measure social support we used a questionnaire with limited ability to differentiate between people with high values in social support. Also, the central assumption of women being more likely to mobilize social support in times of stress and investing more in their social relationships than men [71] has recently been challenged as there is evidence for men also showing an increase in prosocial behavior during stress [72]. It would thus be interesting to investigate the effects of similar interventions in men and to compare them with effects in women.

In summary, our social support stress management in healthy women was successful in terms of feasibility and applicability and had short-term effects on subjectively reported perceived stress. Further evaluation on subjects reporting low levels of supportive social interactions could elucidate if the intervention can sustainably improve social support. Although research has clearly shown that social support has a notable impact on morbidity and mortality, it rests fairly unclear if these effects can be harnessed in a more systematic way.

## Supporting information

**S1 Checklist. CONSORT 2010 checklist of information to include when reporting a randomised trial**∗.
(DOC)

**S1 Appendix.**
(DOCX)

## Acknowledgments

We thank David Bürgin, Felicitas Forrer, Edward Grey, Sebastian Hasler, Stefanie Urech, and Christoph Werner for helping to implement the stress test and collect data.

## Author Contributions

**Conceptualization:** Nadja Heimgartner, Stefanie Grolimund, Svetlana Ponti, Jens Gaab.

**Data curation:** Nadja Heimgartner, Jens Gaab.

**Formal analysis:** Nadja Heimgartner, Sibylle Meier, Jens Gaab.

**Funding acquisition:** Nadja Heimgartner, Jens Gaab.

**Investigation:** Nadja Heimgartner, Stefanie Grolimund, Svetlana Ponti, Silvana Arpagaus, Flurina Kappeler, Jens Gaab.

**Methodology:** Nadja Heimgartner, Sibylle Meier, Stefanie Grolimund, Svetlana Ponti, Silvana Arpagaus, Flurina Kappeler, Jens Gaab.

**Project administration:** Nadja Heimgartner, Sibylle Meier, Stefanie Grolimund, Svetlana Ponti, Silvana Arpagaus, Flurina Kappeler, Jens Gaab.

**Resources:** Nadja Heimgartner, Sibylle Meier, Stefanie Grolimund, Svetlana Ponti, Silvana Arpagaus, Flurina Kappeler.

**Software:** Nadja Heimgartner, Sibylle Meier, Stefanie Grolimund, Svetlana Ponti.

**Supervision:** Nadja Heimgartner, Jens Gaab.

**Validation:** Nadja Heimgartner, Silvana Arpagaus, Flurina Kappeler.

**Visualization:** Nadja Heimgartner, Jens Gaab.

**Writing – original draft:** Nadja Heimgartner, Jens Gaab.

**Writing – review & editing:** Nadja Heimgartner, Sibylle Meier, Jens Gaab.

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
