## [Decision Letter · Decision Letter 0]

15 May 2020

PONE-D-20-02141

Randomized controlled evaluation of the psychophysiological effects of social support stress management in healthy women

PLOS ONE

Dear Prof. Dr. Gaab,

Thank you for submitting your manuscript to PLOS ONE. After careful consideration, we feel that it has merit but does not fully meet PLOS ONE’s publication criteria as it currently stands. Therefore, we invite you to submit a revised version of the manuscript that addresses the points raised during the review process.

Please be aware that reviewers raised substantial concerns, and a complete revision answering all queries point-by-point is mandatory for further assessement. Especially referees #2 and #3 question your statistics and design, points that need to be properly addressed.

We would appreciate receiving your revised manuscript by Jun 27 2020 11:59PM. To enhance the reproducibility of your results, we recommend that if applicable you deposit your laboratory protocols in protocols.io, where a protocol can be assigned its own identifier (DOI) such that it can be cited independently in the future. For instructions see: http://journals.plos.org/plosone/s/submission-guidelines#loc-laboratory-protocols

We look forward to receiving your revised manuscript.

Kind regards,

Johannes Fleckenstein

Academic Editor

PLOS ONE

Journal Requirements:

2. Thank you for submitting your clinical trial to PLOS ONE and for providing the name of the registry and the registration number. The information in the registry entry suggests that your trial was registered after patient recruitment began. PLOS ONE strongly encourages authors to register all trials before recruiting the first participant in a study.As per the journal’s editorial policy, please include in the Methods section of your paper:

i) your reasons for your delay in registering this study (after enrolment of participants started);

ii) confirmation that all related trials are registered by stating: “The authors confirm that all ongoing and related trials for this drug/intervention are registered”.Please also ensure you report the date at which the ethics committee approved the study as well as the complete date range for patient recruitment and follow-up in the Methods section of your manuscript.               

3. Please include a your captions for each figure within your manuscript.

4. Please include your tables as part of your main manuscript and remove the individual files. Please note that supplementary tables (should remain/ be uploaded) as separate "supporting information" files.

Reviewers' comments:

Reviewer's Responses to Questions

**Comments to the Author**

1. Is the manuscript technically sound, and do the data support the conclusions?

Reviewer #1: Yes

Reviewer #2: Partly

Reviewer #3: No

2. Has the statistical analysis been performed appropriately and rigorously? 

Reviewer #1: Yes

Reviewer #2: No

Reviewer #3: N/A

3. Have the authors made all data underlying the findings in their manuscript fully available?

Reviewer #1: Yes

Reviewer #2: No

Reviewer #3: No

4. Is the manuscript presented in an intelligible fashion and written in standard English?

Reviewer #1: No

Reviewer #2: Yes

Reviewer #3: No

5. Review Comments to the Author

Reviewer #1: There are minor errors in citation, typography and grammar that need to be addressed:

- page 3: "a survey in 2012 showed that ... (American Psychological Association, 2012)" the source is not listed in the References. Besides it would be helpful for the reader to know, in which country this survey has been conducted. Please check all references for accuracy and completeness.

- page 3: please change "psychoscial stressor" into "psychosocial stressor"

- page 4: please change "effects of a intervention" into "effects of an intervention"

- page 10: please change "Qui-square tests" into "Chi-square tests"

- page 14: please change "although this studies" into "although these studies"

- page 15: please change "... indicating no effect of the intervention on acute stress responses." into "indicating no effect of the intervention on acute physiological stress response."

other comments:

In the discussion section the following is stated: (page 15) "With regard to our intervention, this finding [i.e. intervention based on social support failed to elicit effects on the cardiovascular stress response] is possible due to the protocol of the TSST, which clearly prevents any possibility to use or obtain social support, as the audience of confederates is instructed to interact in a neutral way with the participants and not to give any positive verbal or non-verbal feedback or signs of communication other than that specified in the manual (Kirschbaum et al., 1993)."

Is this really so? What about accessing internalized social support? In fact, in the Appendix A (page 26) the authors themselves declare that "the use of imagined persons as social support in times when no-one is directly available" was a key topic in the 5th session of their social support mangangement intervention. I suggest the authors to discuss the relevance of further investigations regarding the stress buffering potential of social support in the absence of a support giver and/or receiver, e.g. imagined, internalized or even embodied social support, or even "inner child" work.

Figure 1: Recruitment and participants flow through the study.

- If possible, please specify the reasons for declined participation after randomization.

- If possible, please specify the reasons for insufficient training attendance.

Reviewer #2: This is a well-presented paper that reported a randomised controlled trial of 53 healthy women who were randomly allocated to receive a two-week social support stress management intervention or a waitlist control condition. Assessments were conducted at baseline, post-intervention and follow-up four weeks later.

As the authors pointed out in the discussion, the total sample size was rather small which limited the statistical power to detect small but clinically relevant differences between two conditions. The assumptions used for sample size calculation as reported in the paper, were also different from those stated in the original study protocol. Was this a post-hoc power calculation? Without a clear definition of the primary outcome, which is compulsory for a randomised controlled trial, it is hard to judge whether the calculated sample size was indeed adequate.

The principle of ITT is to include all randomised participants (i.e. 28 intervention and 25 control condition), regardless of the treatment they actually received during the trial. The authors did not define the ITT population used for the trial, and only included 20 participants in each condition (Figure 1 and Table 1). As reported in the literature, the last value carried forward approach is simple to administer but can give a biased estimate of the treatment effect and underestimate the variability of the estimated result. A more robust approach is to use multiple imputations that account for imputation uncertainty, or advanced mixed models with repeated measures. Regardless, the amount of missing data reported in the study is of concern and the results based on a further reduced sample size must be interpreted with caution.

Baseline imbalance was tested between two conditions, although any observed differences between groups should only have occurred by chance if the randomisation was implemented properly. Please provide more details on how the randomisation list was generated and by whom.

How was the effect size quantified and for which time point? What was the rationale to measure the outcomes at both post-intervention and follow-up? For repeated measures analysis, the time point should be fitted in the model as well as its interaction with the randomised condition so that the group difference could be tested at each time point. Alternatively, the authors could run separate regression analysis at each of the two visits post randomisation and report the estimated effect sizes with associated 95% confidence intervals. Note that F-statistics and p-values alone provide no information on the size of intervention effect. For all outcome measures, please explain the range of each calculated scale and how it is interpreted. Table 1 should report the mean and standard deviation (SD) rather than SEM. Please also follow the suggested table templates in the CONSORT 2010 statements and report all outcome measures in the table with descriptive statistics and estimated group differences with 95% CIs.

Reviewer #3: General comments

The objective of the article is to assess the effect of an intervention focused on training interpersonal skills under

different dimensions of the stress response. In terms of the experimental design, the study has issues that must be

clarified. Among others, it would have been good to implement a pre-post design and thus rule out individual

differences in the stress response. With respect to the impact of the study, the scope and novelty of the (only)

positive result report is not completely clear. Indeed, the authors of the article recognize existing evidence in this

direction (e.g., Gaab et al., 2003; Storch et al., 2007; Martín et al., 2011).

Ultimately, the experiment does not present results in the physiological dimension, but rather in a self-reporting measurement, which is to be expected and not entirely relevant (e.g., Reyes et al. 2005). In fact, reporting the belief of a “stress reduction” does not seem to be connected to physiological changes, which is why it does not seem correct to conclude the article in terms of stress as such. To sum up and considering all the aspects indicated, I believe it is an initial

investigation, a first experimental approach that needs to be redesigned. I recommend that the article be re-

submitted after correcting the experimental design.

Specific points

1/ present Appendix 1 in the main text.

2/ I do not fully understand if the knowledge of the participants regarding the intervention could explain the

effect evidenced (on stress perception, PSS).

3/ If the study tries to insist on the importance of training in interpersonal skills, the results are not in line with

this objective. At least this point must be addressed in more detail in the discussion. 

4/ The authors maintain: “The TSST elicited an increase in cortisol, heart rate and state anxiety and a decrease

in RMSSD”. Indeed, the authors of the article recognize existing evidence in this direction (e.g., Gaab et al.,

2003; Storch et al., 2007; Martín et al., 2011).However, this statement is not statistically justified in the text.

5/ It is recommended that the writing and format be revised in detail.

6. PLOS authors have the option to publish the peer review history of their article (what does this mean?). If published, this will include your full peer review and any attached files.

Reviewer #1: Yes: Marko Nedeljkovic

Reviewer #2: No

Reviewer #3: No

---

## [Author Response · Author response to Decision Letter 0]

12 Mar 2021

Response to Reviewers

Responses to the academic editor 

Thank you very much for your feedback. Please excuse the very late resubmission of the manuscript, which was caused by the impact of the Covid pandemic. Our lab was strongly affected and the workflow was less than optimal. Please find our responses and actions to your comments in the following:

1. The manuscript is now formatted to meet PLOS ONE style requirements. 

2. Your comment: The information in the registry entry suggests that your trial was registered after patient recruitment began. PLOS ONE strongly encourages authors to register all trials before recruiting the first participant in a study. As per the journal’s editorial policy, please include in the Methods section of your paper:

i) your reasons for your delay in registering this study (after enrolment of participants started)

ii) confirmation that all related trials are registered by stating: “The authors confirm that all ongoing and related trials for this drug/intervention are registered”.Please also ensure you report the date at which the ethics committee approved the study as well as the complete date range for patient recruitment and follow-up in the Methods section of your manuscript.

At the time we started the recruitment for the study, it was not common use in Switzerland to register trials with non-clinical populations. However, due to experiences with PLOS One (Tondorf et al. 2017, Employing open/hidden administration in psychotherapy research: A randomized-controlled trial of expressive writing), we registered the study. A confirmation that all related trials are registered, the date of the ethics approval and the complete data range (recruitment to follow-up) are added.

3. Captions for figures are added. 

4. The table is now included as part of the manuscript. As reviewer 2 asked to report all outcome measures in the table with descriptive statistics and estimated group differences with 95% CIs, these are added to table 1 and saved in the document “Table 1 supporting information”. Due to the bigger format, this table could not be included in the manuscript. 

Your comment: We note that you have indicated that data from this study are available upon request. PLOS only allows data to be available upon request if there are legal or ethical restrictions on sharing data publicly. For more information on unacceptable data access restrictions, please see http://journals.plos.org/plosone/s/data-availability#loc-unacceptable-data-access-restrictions.

According to the Swiss regulation on human research we are not allowed to publish data from a clinical trial on a public repository. The department of clinical research at the university of Basel set up a data access committee, which publishes data in form of meta data on a public repository and answers to requests from researchers to get access to the original data. We will provide you with the information regarding the access of our metadata within the next two weeks. 

 

Responses to reviewer #1

Thank you very much for your thorough and most helpful feedback. Please excuse the very late resubmission of the manuscript, which was caused by the impact of the Covid pandemic. Our lab was strongly affected and the workflow was less than optimal. Please find our responses and actions to your comments in the following:

• Your comment: There are minor errors in citation, typography and grammar that need to be addressed: - page 3: "a survey in 2012 showed that ... (American Psychological Association, 2012)" the source is not listed in the References. Besides it would be helpful for the reader to know, in which country this survey has been conducted. Please check all references for accuracy and completeness. - page 3: please change "psychoscial stressor" into "psychosocial stressor" - page 4: please change "effects of a intervention" into "effects of an intervention" - page 10: please change "Qui-square tests" into "Chi-square tests" - page 14: please change "although this studies" into "although these studies" - page 15: please change "... indicating no effect of the intervention on acute stress responses." into "indicating no effect of the intervention on acute physiological stress response."

The errors in citation, typography, and grammar have been addressed in the revisited manuscript. 

• Your comment: In the discussion section the following is stated: (page 15) "With regard to our intervention, this finding [i.e. intervention based on social support failed to elicit effects on the cardiovascular stress response] is possible due to the protocol of the TSST, which clearly prevents any possibility to use or obtain social support, as the audience of confederates is instructed to interact in a neutral way with the participants and not to give any positive verbal or non-verbal feedback or signs of communication other than that specified in the manual (Kirschbaum et al., 1993)."  Is this really so? What about accessing internalized social support? In fact, in the Appendix A (page 26) the authors themselves declare that "the use of imagined persons as social support in times when no-one is directly available" was a key topic in the 5th session of their social support mangangement intervention. I suggest the authors to discuss the relevance of further investigations regarding the stress buffering potential of social support in the absence of a support giver and/or receiver, e.g. imagined, internalized or even embodied social support, or even "inner child" work.

We also addressed your thoughts on the role of internalized social support during the TSST in the discussion. 

• Your comment: Figure 1: Recruitment and participants flow through the study. - If possible, please specify the reasons for declined participation after randomization. - If possible, please specify the reasons for insufficient training attendance.

Of the six participants withdrawing after allocations, only two cancelled their participation via email due to lack of interest to continue any further. Reasons for insufficient intervention attendance were not asked for. For the revised manuscript, we repeated the analysis of the perceived stress data including the five subjects with insufficient intervention attendance, which still led to a significant difference between conditions at post-intervention.

 

Responses to reviewer #2

Thank you very much for your thoughtful and most helpful feedback, especially on the handling of missing data and choice of statistical models to test for intervention effects. Please excuse the very late resubmission of the manuscript, which was caused by the impact of the Covid pandemic. Our lab was strongly affected and the workflow was less than optimal. Please find our responses and actions to your comments in the following:

• Your comment: As the authors pointed out in the discussion, the total sample size was rather small which limited the statistical power to detect small but clinically relevant differences between two conditions. The assumptions used for sample size calculation as reported in the paper, were also different from those stated in the original study protocol. Was this a post-hoc power calculation? Without a clear definition of the primary outcome, which is compulsory for a randomised controlled trial, it is hard to judge whether the calculated sample size was indeed adequate.

Regarding the sample size calculations, we used the same approach as in our previous papers on this matter (Gaab et al. 2003, Randomized controlled evaluation of the effects of cognitive–behavioral stress management on cortisol responses to acute stress in healthy subjects; Hammerfald et al. 2006, Persistent effects of cognitive-behavioral stress management on cortisol responses to acute stress in healthy subjects—a randomized controlled trial, Storch et al. 2007, Psychoneuroendocrine effects of resource-activating stress management training). The primary outome was the cortisdol stress reactivity and in the aforementioned publications, this sample size was well-suited to detect differences in cortisol stress reactivity between two groups. However, we had a drop-out of N=6 after randomization and another drop out of N=5 for the Trier Social Stress Test, resulting in a N=37 in the cortisol sample. Thus, we can’t exclude that detecting a difference between conditions was not possible due to underpowering. However, examining the results of the cortisol stress response (Figure 2) does not indicate that more participants would have changed results in physiological stress reactions significantly. We adressed his also in the discussion section. 

• Your comment: The principle of ITT is to include all randomised participants (i.e. 28 intervention and 25 control condition), regardless of the treatment they actually received during the trial. The authors did not define the ITT population used for the trial, and only included 20 participants in each condition (Figure 1 and Table 1). As reported in the literature, the last value carried forward approach is simple to administer but can give a biased estimate of the treatment effect and underestimate the variability of the estimated result. A more robust approach is to use multiple imputations that account for imputation uncertainty, or advanced mixed models with repeated measures. Regardless, the amount of missing data reported in the study is of concern and the results based on a further reduced sample size must be interpreted with caution.

Based on your comments, we re-analyzed all data using covariance pattern models with repeated measures (MIXED procedure in SPSS) so we could include all the subjects with at least on measurement in the repeatedly measured variable. To avoid underestimation of standard errors and also the loss of the relationship between variables, we choose to not impute data sets when all measurements of the interested parameter were missing. Please find the new analysis with adapted sample sizes in the results section. Additionally, to be able to test if the intervention had an effect, subjects of the intervention sample had to attend at least eight hours of the intervention. Perceived stress was the only outcome where we could find a difference between conditions over the assessments. We repeated the analysis for all subjects including the ones with six and less hours of training attendance and could still find the difference at post-intervention. 

• Your comment: Baseline imbalance was tested between two conditions, although any observed differences between groups should only have occurred by chance if the randomisation was implemented properly. Please provide more details on how the randomisation list was generated and by whom.

Regarding the randomization method, please find more detailed information in the methods section. 

• Your comment: How was the effect size quantified and for which time point? What was the rationale to measure the outcomes at both post-intervention and follow-up? For repeated measures analysis, the time point should be fitted in the model as well as its interaction with the randomised condition so that the group difference could be tested at each time point. Alternatively, the authors could run separate regression analysis at each of the two visits post randomisation and report the estimated effect sizes with associated 95% confidence intervals. Note that F-statistics and p-values alone provide no information on the size of intervention effect. For all outcome measures, please explain the range of each calculated scale and how it is interpreted. Table 1 should report the mean and standard deviation (SD) rather than SEM. Please also follow the suggested table templates in the CONSORT 2010 statements and report all outcome measures in the table with descriptive statistics and estimated group differences with 95% CIs.

For models where the interaction of condition by time was significant, we included unstandardized regression coefficients with confidence intervals to show the effects of condition at post-intervention and follow-up. Assessments four-weeks later were included in the models to test if the possible effect of the intervention was sustainable over a longer period of time or just a direct short effect of the intervention. Please find descriptive statistics and estimated group differences with 95% CIs in the additional “table 1 supporting information. 

 

Responses to reviewer #3

Thank you for your thoughtful and most helpful feedback to our study. Please excuse the very late resubmission of the manuscript, which was caused by the impact of the Covid pandemic. Our lab was strongly affected and the workflow was less than optimal. Please find our responses and actions to your comments in the following:

• Your comment: The objective of the article is to assess the effect of an intervention focused on training interpersonal skills under different dimensions of the stress response. In terms of the experimental design, the study has issues that must be clarified. Among others, it would have been good to implement a pre-post design and thus rule out individual differences in the stress response. With respect to the impact of the study, the scope and novelty of the (only) positive result report is not completely clear. Indeed, the authors of the article recognize existing evidence in this direction (e.g., Gaab et al., 2003; Storch et al., 2007; Martín et al., 2011).

You argued for an implementation of a pre-post design to test for effects of the intervention on the acute stress reaction. Unfortunately, it is explicitly recommended to apply the Trier Social Stress Test only once to the same participants as studies showed a high degree of habituation of the HPA axis response to the TSST with repeated exposures (see Pruessner, Gaab, Hellhammer, Lintz, Schommer & Kirschbaum, 1997; Schommer, Hellhammer & Kirschbaum, 2003). To our knowledge, there exists no other well evaluated psychosocial stress test today which is not susceptible to such effects of adaptation. Therefore, we measured the reaction to acute psychosocial stress only once, after the intervention took place, and tried to control for known factors that influence the stress reaction by randomizing the participants to the two conditions and evaluating known factors that can influence the stress reaction (smoking, age, BMI, hormonal contraceptives, menstruation phase). 

Regarding the novelty of our approach, we set out to use an established methodological approach to test a novel intervention. Thus, in order to examine possible effects of a novel stress management training, i. e. based on the principles of social support, we used the the same approach as in a number of published publications of our group (Gaab, Blättler, Menzi, Pabst, Stoyer, Ehlert. Randomized controlled evaluation of the effects of cognitive–behavioral stress management on cortisol responses to acute stress in healthy subjects. Psychoneuroendocrinology. 2003;28(6):767-79; Hammerfald, Eberle, Grau, Kinsperger, Zimmermann, Ehlert, et al. Persistent effects of cognitive-behavioral stress management on cortisol responses to acute stress in healthy subjects—a randomized controlled trial. Psychoneuroendocrinology. 2006;31(3):333-9; Storch, Gaab, Küttel, Stüssi, Fend. Psychoneuroendocrine effects of resource-activating stress management training. Health Psychol. 2007;26(4):456-63). With this approach, we were able to have a valid and reliable test of our intervention and also, allow comparability to these published studies. With regard to our intervention, the only comparable pulication (Anthony and O’Brien, 2003) also used an intervention based on social support. However, the intervention of Anthony and O’Brien was very brief and thus very unlikely to result in any effect. In fact, the intervention was only for one hour and they also did not assess endocrine responses to acute stress. 

• Your comment: Ultimately, the experiment does not present results in the physiological dimension, but rather in a self-reporting measurement, which is to be expected and not entirely relevant (e.g., Reyes et al. 2005). In fact, reporting the belief of a “stress reduction” does not seem to be connected to physiological changes, which is why it does not seem correct to conclude the article in terms of stress as such. To sum up and considering all the aspects indicated, I believe it is an initial investigation, a first experimental approach that needs to be redesigned. I recommend that the article be re- submitted after correcting the experimental design.

Stress is a multidimensional phenomenon and thus, it is not possible to play out one dimension against another. We assessed both physiological and psychological dimensions of stress and our findings indicate that the training had no effects on acute physiological and emotional stress responses, but that perceived stress in general was reduced. The methodological design and experimental setup were used in a number of published studies on this matter (see above) and thus, we are convinced that our approach is valid and established.

1) Your comment: present Appendix 1 in the main text.

We included the appendix in the method section

2) Your comment: I do not fully understand if the knowledge of the participants regarding the intervention could explain the effect evidenced (on stress perception, PSS).

We agree that it might be possible that participants knowledge about the intervention could be an explanation of the observed effect. As with most psychological interventions, participants in our training were not blind to their allocation and thus were fully aware that they are receiving a possibly effective treatment. This knowledge might have triggered a response expectancy, which in turn is known to be related to treatment outcome. We have included this in the discussion section.

3) Your comment: If the study tries to insist on the importance of training in interpersonal skills, the results are not in line with this objective. At least this point must be addressed in more detail in the discussion. 

We agree that the question if social support is amenable to training is a valid one and we included a paragraph on this in the discussion section.

4) Your comment: The authors maintain: “The TSST elicited an increase in cortisol, heart rate and state anxiety and a decrease in RMSSD”. Indeed, the authors of the article recognize existing evidence in this direction (e.g., Gaab et al., 2003; Storch et al., 2007; Martín et al., 2011). However, this statement is not statistically justified in the text.

We included the statistics for the time effects of the variables measured before, during and after the TSST in the result section.

5) Your comment: It is recommended that the writing and format be revised in detail.

We thoroughly checked and changed the text in format and writing.

---

## [Decision Letter · Decision Letter 1]

19 May 2021

Randomized controlled evaluation of the psychophysiological effects of social support stress management in healthy women

PONE-D-20-02141R1

Dear Dr. Gaab,

We’re pleased to inform you that your manuscript has been judged scientifically suitable for publication and will be formally accepted for publication once it meets all outstanding technical requirements.

Kind regards,

Johannes Fleckenstein

Academic Editor

PLOS ONE

Additional Editor Comments (optional):

As you will see below, Reviewer 3 has outstanding concerns with the manuscript. However, having evaluated the manuscript myself I do not consider any further revisions to be necessary.

Reviewers' comments:

Reviewer's Responses to Questions

**Comments to the Author**

1. If the authors have adequately addressed your comments raised in a previous round of review and you feel that this manuscript is now acceptable for publication, you may indicate that here to bypass the “Comments to the Author” section, enter your conflict of interest statement in the “Confidential to Editor” section, and submit your "Accept" recommendation.

Reviewer #1: All comments have been addressed

Reviewer #3: All comments have been addressed

2. Is the manuscript technically sound, and do the data support the conclusions?

Reviewer #1: Yes

Reviewer #3: Partly

3. Has the statistical analysis been performed appropriately and rigorously? 

Reviewer #1: Yes

Reviewer #3: Yes

4. Have the authors made all data underlying the findings in their manuscript fully available?

Reviewer #1: Yes

Reviewer #3: Yes

5. Is the manuscript presented in an intelligible fashion and written in standard English?

Reviewer #1: Yes

Reviewer #3: Yes

6. Review Comments to the Author

Reviewer #1: Thank you for thoroughly addressing all my previous comments. From my point of view, there are no further ammendments pending.

Reviewer #3: The aim of the article is to examine the effect of a social support stress management intervention on psychophysiological stress responses as well as self-reported stress in healthy women. Results suggest that he intervention had no effect on the physiological (stress) measure across groups (HRV, Cortisol concentration), but only a small effect on self-reported stress.

In general terms, it seems to be a well-developed clinical trial with an adequate control of the psychological and physiological indicators. However, unfortunately, I do not recommend the publication, mainly due to the lack of clarity regarding the stress effect reported. Ultimately, the authors should clarify / justify / argue why the null results in physiology might not be relevant for the type of intervention implemented. And from there, why the small effect on the self-report of stress could be relevant, when it does not seem to persist in the follow-up stage. Even if it could be the case that the intervention effect is presented exclusively at the self-report level, the authors should explain if such effect is not simply due to their knowledge of the experimental protocol (i.e., the “discuss of the intervention” explicit on the first session).

Our suggestion is that the team of authors carry out a new and detailed review of the experimental protocol. If the reported stress effect is consistent, then a replication of this effect is desirable, experimentally controlling the knowledge that the participants have of the intervention. Associated with the above, the authors should better argue why the stress effect is (and should be in this type of intervention) independent of the expected effect on physiology.

7. PLOS authors have the option to publish the peer review history of their article (what does this mean?). If published, this will include your full peer review and any attached files.

Reviewer #1: No

Reviewer #3: **Yes: **Gabriel Reyes M

---

## [Editor Report · Acceptance letter]

26 May 2021

PONE-D-20-02141R1 

Randomized controlled evaluation of the psychophysiological effects of social support stress management in healthy women 

Dear Dr. Heimgartner:

I'm pleased to inform you that your manuscript has been deemed suitable for publication in PLOS ONE. Congratulations! Your manuscript is now with our production department. 

Kind regards, 

on behalf of

Priv.-Doz. Dr. Johannes Fleckenstein 

Academic Editor

PLOS ONE